# An Ultra-Micro-Volume Adhesive Transfer Method and Its Application in fL–pL-Level Adhesive Distribution

**DOI:** 10.3390/mi13050664

**Published:** 2022-04-23

**Authors:** Huifang Liu, Xi Chen, Shuqing Wang, Shenhui Jiang, Ying Chen, Fuxuan Li

**Affiliations:** 1School of Mechanical Engineering, Shenyang University of Technology, Shenyang 110870, China; chenxi897918099@163.com (X.C.); jshenhui0204@163.com (S.J.); chen1357928@163.com (Y.C.); lfx_19961213@163.com (F.L.); 2Shenyang Machine Tool (Group) Co., Ltd., Shenyang 110142, China; shuqing_wang@126.com

**Keywords:** fL–pL level, transfer droplet volume, microporous encapsulation, ultra-micro-dispensing

## Abstract

This study is aimed at addressing the urgent demand for ultra-micro-precision dispensing technology in high-performance micro- and nanometer encapsulation, connection, and assembly manufacturing, considering the great influence of colloid viscosity and surface tension on the dispensing process in micro- and nanometer scale. According to the principle of liquid transfer, a method of adhesive transfer that can realize fL–pL levels is studied in this paper. A mathematical model describing the initial droplet volume and the transfer droplet volume was established, and the factors affecting the transfer process of adhesive were analyzed by the model. The theoretical model of the transfer droplet volume was verified by a 3D scanning method. The relationships between the transfer droplet volume and the initial droplet volume, stay time, initial distance, and stretching speed were systematically analyzed by a single-factor experiment, and the adhesive transfer rate was calculated. Combined with trajectory planning, continuous automatic dispensing experiments with different patterns were developed, and the problems of the transfer droplet size, appearance quality, and position accuracy were analyzed comprehensively. The results show that the average relative deviation of the transfer droplet lattice position obtained by the dispensing method in this paper was 6.2%. The minimum radius of the transfer droplet was 11.7 μm, and the minimum volume of the transfer droplet was 573.3 fL. Furthermore, microporous encapsulation was realized using the method of ultra-micro-dispensing.

## 1. Introduction

MEMS technology is currently important for improving aerospace and military capabilities [1]. Although the development of micro- and nanometer devices is relatively mature, many microelectromechanical systems cannot be applied in practice. Difficulties are related to the high-quality encapsulation, connection, assembly, and integration technology of micro- and nanometer devices. Adhesive micro-distribution technology is an important enabling core technology for the encapsulation, connection, and integration of micro- and nanometer devices. Adhesive micro-distribution technology refers to the use of adhesives to connect micro-sized components [2]. Common dispensing methods include the time/pressure type, piston pump type, transfer printing type, piezoelectric stack type, and piezoelectric ceramic type. Shi proposed a time/pressure-type pL-level micro-volume adhesive transfer method with a minimum dispensing volume of 2 pL, but the viscosity of the applicable adhesive was low [3]. Chen designed a piston pump-type dispensing system with a minimum dispensing volume of 50 μL and a minimum controllable increment of 0.1 μL [4]. Zhang proposed a transfer printing-type dispensing pen. It achieved microporous sealing of 5–20 μm, and the minimum dispensing volume was 4.4 pL [5]. Gu proposed a piezoelectric stack-type adhesive spray system that could spray 143 transfer droplets per second; the droplet volume error was within 3.11%, and the droplet average volume was 0.564 μL [6]. Fan proposed a method to squeeze a capillary tube through a piezoelectric ceramic tube for adhesive transfer, achieving pL-level low-viscosity adhesive transfer, and the volume of the smallest transfer droplet was 8.31 pL [7]. Compared with welding, thermal bonding, and electrostatic bonding, this method does not require high-temperature and high-pressure conditions, while it has the advantages of simple implementation and low stress [8,9,10]. Therefore, it is widely used in the assembly of microsystems [11]. For example, adhesive transfer technology enables the surface encapsulation of a micro-level electrostatic motor. The micro-accelerometer, which is installed in the automobile airbag, plays a crucial role in driving safety. Using adhesive bonding technology to fix and encapsulate the micro-accelerometer can improve the stability of the device and prolong its service life, thus making driving safer. The adhesive transfer technology proposed in this paper can achieve precise control at the micron level, with a minimum dispensing volume reaching the fL level and the potential for encapsulation in a narrow space. Therefore, it is very advantageous to use this system to fix and encapsulate micro-electrostatic motors and micro-accelerometers. However, too much adhesive at the connection will lead to overflow and pollution, while insufficient adhesive will lead to a decline in connection strength [12]. Obviously, a precise control adhesive transfer volume is very important for the encapsulation, connection, assembly, and integration of micro- and nanometer devices. When using noncontact adhesive spraying for liquid transfer, the inner diameter of the hollow needle must be reduced to a few microns or even smaller, which is very difficult to manufacture [13,14]. Therefore, there are still many problems to be solved in the application of jet methods to adhesive transfer with fL–pL resolution. With the development of MEMS technology, the original liquid transfer technology cannot meet the precision requirements of micro-connections. Therefore, there is an urgent need for an ultra-micro-automatic dispensing technology matching the size and accuracy of micro- and nanometer manufacturing.

The method of transfer printing is to impregnate the liquid through the stamp, and then transfer the impregnated liquid to the base surface through contact. Because the method of transfer printing is based on liquid surface flow rather than internal pipe flow, the effect of high flow resistance on liquid transfer can be reduced, which is promising in the field of micro-volume liquid transfer. Microarray transfer printing technology has been used to generate micro-patterns of various biomolecules and photoresists [15]. The pioneering studies of liquid transfer were conducted by Chadov and Yakhnin [16]. They mainly studied the situation of liquid transfer at different stretching speeds. Three states of liquid transfer were defined, namely, a quasi-static regime, dynamic regime, and transition regime. In addition, Cai’s team studied the effect of initial distance on the liquid transfer. They found that, in the process of liquid transfer, when the distance between the donor surface and acceptor surface decreases, the contact area between the liquid and acceptor surface increases [17,18,19]. Dodds et al. studied the effect of different contact angles on the liquid transfer process [20]. The research showed that there is a lag phenomenon of contact angle in the liquid transfer process, and that the contact angle has an important influence on the liquid transfer volume [21,22,23,24,25,26]. On the basis of the existing literature, it can be found that most studies did not comprehensively evaluate the relationships between each physical parameter and the adhesive transfer volume. The adhesive transfer volume described in most studies could only reach sub-nanoliter levels, but not fL–pL levels.

This study is aimed at addressing the urgent need for high-precision adhesive transfer technology in the micro- and nanometer encapsulation industry, considering the influence of the adhesive properties on the transfer process, as well as the problem of micro-nozzles being difficult to manufacture. On the basis of the principle of liquid transfer, an adhesive transfer method which can realize fL–pL levels is studied in this paper. A mathematical model describing the initial droplet volume and the transfer droplet volume is established, and the factors affecting the transfer process of adhesive are analyzed by the model. The relationships between the transfer droplet volume and initial droplet volume, stay time, initial distance, and stretching speed are analyzed using a single-factor method. Furthermore, combined with trajectory planning, continuous automatic dispensing experiments with different patterns are carried out, and the research results are applied in microporous encapsulation.

## 2. Principle of Adhesive Transfer and Modeling of Transfer Quantity

### 2.1. Adhesive Transfer Principle

The adhesive transfer principle proposed in this paper is to attach the adhesive to the pipetting needle first, and then control the pipetting needle attached with the adhesive to contact the specified base surface. The adhesive is partly transferred from the pipetting needle to the base surface by relying on its own viscosity, and the transferred adhesive forms transfer droplets under the action of surface tension.

The ultra-micro-dispensing process based on the transfer principle is shown in Figure 1. The dispensing process is divided into five stages: adhesive taking, transportation, extrusion, stretching, and return. In the adhesive taking stage, the pipetting needle moves downward within the capillary tube fitted with the adhesive, as shown in Figure 1a. In the transport stage, the pipetting needle passes through the capillary tube. Under the action of viscous force, the adhesive adheres to the surface of the pipetting needle. Under the action of surface tension, the adhesive partly converges at the tip of the pipetting needle to form initial droplets. The pipetting needle with adhesive continues to move downward, as shown in Figure 1b. When the adhesive comes into contact with the base surface, the extrusion stage begins. At this time, the adhesive connects the pipetting needle with the base surface to form a liquid bridge. The liquid bridge is squeezed until the preset initial distance threshold is reached, and the extrusion process ends, as shown in Figure 1c. In the stretching stage, the pipetting needle moves upward, which pulls the liquid bridge off. Some of the adhesive in contact with the liquid bridge is transferred to the base surface, and the transferred adhesive forms transfer droplets under the combined action of surface tension and viscous force, as shown in Figure 1d,e. In the return stage, the pipetting needle moves upward, and the adhesive at the capillary tube mouth moves upward together with the pipetting needle under the combined action of inertial force, viscous force, and surface tension to form a liquid fossa. When the height of the pipetting needle returning to the capillary tube is greater than the height of the liquid fossa, the return stage is over and the system completes an adhesive distribution, as shown in Figure 1f.

### 2.2. Adhesive Transfer Rate Modeling

In the dispensing process, the adhesive adheres to the surface of the pipetting needle. Through extrusion and stretching, some of the adhesive on the pipetting needle is transferred to the base surface to form transfer droplets, and the residual adhesive on the pipetting needle is brought back to the capillary tube. In this process, the adhesive initially adhered to the surface of the pipetting needle is not completely transferred to the base surface. In order to study the law of adhesive transfer, it is necessary to theoretically calculate and analyze the adhesive transfer rate.

After the pipetting needle passes through the capillary tube, the adhesive adheres to the surface of the pipetting needle under the combined action of viscous force and surface tension. Assuming that the state between the pipetting needle and the adhesive is ideal, the section passing through the axis of the pipetting needle was taken as the research object, as shown in Figure 2.

In Figure 2, the diameter of the pipetting needle is *d*. The thickness of the adhesive adhering to the surface of the pipetting needle is *δ*. The curved surface radius of the initial droplet is *r*_1_. The contact angle formed between the tip of the pipetting needle and initial droplet is *θ*. The difference between the circle A’s radius and the initial droplet height is *a*. The intersection formed by the straight line passing through the circle center O_1_ and the tangent line passing through circle A is (*x*_1_, *y*_1_). Taking the *y*-axis as the rotation axis, we can get the initial droplet volume *V*_1_ as follows:(1)$$V_{1}=\int_{a}^{r_{1}}\pi x^{2}dy=\int_{\cot\theta (\frac{d+2\delta }{2})}^{\csc\theta (\frac{d+2\delta }{2})}\pi (r_{1}^{2}-y_{1}^{2})dy.$$

By integrating Equation (1), the initial droplet volume *V*_1_ can be obtained as follows:(2)$$V_{1}=\frac{\pi {\left(d+2\delta \right)}^{3}(2+\cos^{3}\theta -3\cos\theta )}{24\sin^{3}\theta }.$$

When the adhesive at the tip of the pipetting needle contacts the base surface, a liquid bridge is formed due to the combined action of the inertial force and surface tension of the adhesive itself. When the pipetting needle moves upward, the liquid bridge is pulled off due to the viscous force of the adhesive. The adhesive left on the base surface is mainly affected by the combined action of mass force, surface tension, and viscous force. Mass force diffuses the adhesive, while surface tension shrinks the adhesive inward. Viscous force holds the adhesive to the base surface. After a period of time, the force of the adhesive reaches equilibrium, and a transfer droplet is finally formed. In an ideal state, the transfer droplet features a spherical crown, as shown in Figure 3.

In Figure 3, the radius of the transfer droplet is *b*, the height of the transfer droplet is *c*, the coordinates at the boundary intersection of three phases are (*x*_2_, *y*_2_), the contact angle of the transfer droplet with the base surface is *β*, and the curved surface radius of the transfer droplet is *r*_2_. Taking the *y*-axis as the rotation axis, we can determine the transfer droplet volume *V*_2_ as follows:(3)$$V_{2}=\int_{r_{2}-c}^{r_{2}}\pi x^{2}dy=\int_{\frac{b}{\sin\beta }-c}^{\frac{b}{\sin\beta }}\pi (r_{2}^{2}-y_{2}^{2})dy.$$

By integrating Equation (3), the transfer droplet volume *V*_2_ can be further obtained as follows:(4)$$V_{2}=\pi c^{2}(\frac{b}{\sin\beta }-\frac{c}{3}).$$

Therefore, the adhesive transfer rate *η* can be calculated as follows:(5)$$\eta =\frac{V_{2}}{V_{1}}=\frac{8c^{2}\sin^{3}\theta (3b-c\sin\beta )}{\sin\beta {\left(d+2\delta \right)}^{3}(2+\cos^{3}\theta -3\cos\theta )}.$$

It can be seen from Equation (2) that the initial droplet volume is related to the diameter of the pipetting needle, the angle between the tip of the pipetting needle and the initial droplet, and the adhesive thickness. In addition, the diameter of the pipetting needle and the thickness of the adhesive are the main factors affecting the volume of the initial droplet. It can be seen from Equation (4) that the volume of the transfer droplet is related to the height of the transfer droplet, the radius of the transfer droplet, and the contact angle between the transfer droplet and the base surface. It can be seen from Equation (5) that the adhesive transfer rate is related to *c*, *θ*, *b*, *β*, *d*, and *δ*. The size of *c* and *β* is mainly related to the wettability of the base surface. The size of *θ* and *δ* is mainly related to the pipetting needle material and the viscosity of adhesive. In the process of stretching the liquid bridge, the contact line between the liquid bridge and the base surface shrinks inward. The dimension of the transfer droplets is largely dependent on the size of the contact line, and the wettability of the base surface directly affects the formation of the contact line. A better base surface wettability results in a smaller distance of the contact line moving inward, and the phenomenon of contact angle hysteresis is more obvious. This means that more adhesive is transferred to the base surface, and a larger transfer droplet formed. If the influence of material properties and adhesive properties on the adhesive transfer rate is not considered, the factors affecting adhesive transfer rate are only the radius of the transfer droplet and the diameter of the pipetting needle. Therefore, this paper only analyzes the effects of the transfer droplet radius and the pipetting needle diameter on the adhesive transfer rate.

## 3. Experimental System and Method

### 3.1. Development of Experimental System

Through the analysis of the dispensing principle and the adhesive transfer process, the main factors affecting the adhesive transfer rate were preliminarily identified. According to the measurement requirements, the dispensing experimental platform was designed. The ultra-micro-precision automatic dispensing system independently developed and designed is shown in Figure 4. The system consists of a transfer droplet morphology observation and data acquisition module, adhesive transfer module, and drive control module.

The function of the transfer droplet morphology observation and data acquisition module involves auxiliary adjustment for the experimental process, as well as completion of the measurement, annotation, and observation of the transfer droplet. The adhesive transfer module is composed of a pipetting needle, a capillary tube, and a base surface. The pipetting needle is used to complete the adhesion and transfer of the adhesive, the capillary tube is used to store the adhesive, and the base surface is the carrier for the transfer adhesive. The function of the drive control module is to control the movement of the pipetting needle. Under the control of the driving device, the pipetting needle can move the minimum distance up to 1 μm.

The prototypes of the developed dispensing system and the built experimental platform are shown in Figure 5. The transfer droplet morphology observation and data acquisition module is composed of a fine-tuning platform, a microscope, a microscope control handle and a microscope display. The fine-tuning platform and the microscope are an integrated structure. The fine-tuning platform can be adjusted by the fine-tuning knob, and the microscope can be adjusted by the microscope control handle. The magnification of the microscope is between 100× and 1000×, which can meet the observation requirements in most cases. The pipetting needle and capillary tube in the adhesive transfer module are fixed on the Z drive. During the experiment, the adhesive adheres to the pipetting needle, and then the adhesive on the pipetting needle is transferred to the base surface to achieve the dispensing process. The drive control module is composed of a computer, a control panel, a drive controller, and an X/Y/Z drive. The computer is responsible for transmitting the control program to the drive controller, and the motion speed of the drive can be set by the computer. The main function of the control panel is to control the movement trajectory of the pipetting needle and set parameters. The drive controller is responsible for transmitting the motion control signal to the X/Y/Z drive. The X/Y/Z drive executes the move command to drive the pipetting needle to the specified position to complete the transfer of adhesive.

### 3.2. Material

The materials needed for the experiment mainly include the pipetting needle, adhesive, base surface, and capillary tube. The pipetting needle is a key component in the dispensing experiment, and its diameter is required to reach tens of microns. On the one hand, it is necessary to ensure that the pipetting needle does not bend during processing. On the other hand, the tip of the pipetting needle is required to be flat and smooth. Therefore, the pipetting needle should have the characteristics of high hardness, corrosion resistance, and wear resistance. A tungsten needle was selected as the raw material in this paper. The machining process of the pipetting needle was divided into two stages: electrolysis and grinding. After electrolysis, the pipetting needle had a stepped shape with a tapered tip. Then, high-mesh sandpaper was used to further grind the tip to achieve a flat and smooth effect. An image of the pipetting needle is shown in Figure 6.

The adhesive transfer method proposed in this paper is suitable for the transfer of adhesives such as epoxy resin, methyl silicone oil, and polyurethane. Because the purpose of the experiment was to study the influence of the initial droplet volume, stay time, initial distance, and stretching speed on the adhesive transfer rate, the composition and viscosity of adhesive were not taken as the research object. However, experiments using adhesives of different composition and viscosity would yield the same conclusion. Therefore, epoxy resin with a viscosity of 1000 cps was taken as the experimental adhesive.

The base surface was thin glass material with a size of 75 × 25 × 1 mm. An ultrasonic cleaning method was used to ensure that the base surface was clean. The adhesive storage tube was a glass capillary tube with an inner diameter of 0.8 mm and a length of 12 cm.

### 3.3. Experimental Conditions

When observing the dispensing process through the microscope, it was found that small vibration had a great impact on the experimental results. Therefore, in order to reduce the impact of vibration on the dispensing process, the experimental platform was built on a shock absorption platform. During the experiment, the room was kept relatively sealed, with no dust indoors and a room temperature of 23–25 °C. The experimental process was consistent with the theoretical process mentioned in Section 2.1. The actual adhesive transfer process observed by the microscope is shown in Figure 7.

## 4. Factors Affecting the Volume of Transfer Droplets and Comprehensive Experiments

The purpose of this paper was to study the automatic distribution method of an adhesive at the fL–pL level, as well as analyze the influence of working parameters on the adhesive transfer process. Therefore, a method for evaluating the volume of the transfer droplet was first established. Then, the single-factor method was used to analyze the influence of each factor on the transfer droplet volume and the adhesive transfer rate. Lastly, combined with the trajectory planning method, an application experiment of automatic adhesive dispensing with fL–pL resolution was completed.

### 4.1. Method for Determining Transfer Droplet Volume

The shape of the transfer droplet was similar to a spherical crown, but it was not a regular spherical crown. Therefore, it was necessary to verify whether Equation (4) could accurately describe the volume of the transfer droplet through experiments.

After analyzing the adhesive transfer rate through the method of theoretical analysis, it was found that the key to studying the adhesive transfer rate was to determine the volume of the transfer droplet. On the one hand, the 3D scanning method could be used to measure the volume of the transfer droplet. This method could obtain an accurate volume, but its efficiency is low. On the other hand, the theoretical calculation method could be used to calculate the transfer droplet volume. This method was more efficient, but its accuracy needed to be verified.

It can be seen from Equation (4) that the volume of the transfer droplet is related to the height of the transfer droplet, the radius of the transfer droplet, and the contact angle between the transfer droplet and the base surface. On the one hand, we used the plane measurement method to measure the above parameters and substituted the measured values into Equation (4) to calculate the transfer droplet volume. The experimental conditions were as follows: a pipetting needle diameter of 60 μm, a stretching speed of 2 mm/s, a stay time of 1 s, and an initial distance of 0 μm. Under these conditions, several dispensing experiments were completed, and 10 transfer droplets were obtained. Figure 8 shows the transfer droplet parameters measured by microscope. The measured height of the transfer droplet was 13 μm, the diameter was 66.5 μm, and the contact angle was 34°. After calculation, we could calculate the transfer droplet volume as 29.3 fL. It can be seen from the figure that the appearance of the transfer droplet was relatively round, and the side of the transfer droplet was similar to a spherical crown. The same method was used to measure the parameters of other transfer droplets, and then the volume of the transfer droplets was calculated. The specific measurement data and calculation results are shown in Table 1. After measurement, it was found that the height of the transfer droplet was 13–15 μm, the radius of the transfer droplet was 31.9–33.3 μm, and the contact angle was 33°–36°. After calculation, we could get that the minimum volume of the transfer droplet was 27.2 pL, the maximum volume was 37.9 pL, and the theoretical average volume of the transfer droplet was 31.5 pL.

On the other hand, we used 3D scanning measurements to determine the actual volume of the transferred droplet. The principle of 3D scanning measurement is to scan the transfer droplet with a microscope to obtain the original point coordinate data. After the original data are processed, the actual transfer droplet volume can be obtained. Because both the transfer droplet and the base surface have light transmittance, serious deletion can appear during direct scanning. Therefore, it was necessary to spray treatment the base surface with a developer before the dispensing experiment. As the diameter of the developer particles was less than 10 μm, they had no significant effect on the morphology of the transfer droplet. After the developer was completely cured, the experiment was started on the base surface. After the experiment, it was necessary to spray the base surface with the developer again. After the developer was cured, the transfer droplets could be scanned by the 3D scanning method.

The volume of the transfer droplet was measured by the 3D scanning method in three steps: scanning of the original transfer droplet, boundary point processing, and error compensation processing. For example, the transfer droplet in Figure 8 was scanned in 3D to obtain the original image of the transfer droplet, as shown in Figure 9a. It can be seen that there was the problem of a fuzzy boundary around the transfer droplets. Therefore, the point processing method was used to process the transfer droplet image, and the exported image after preliminary processing is shown in Figure 9b. It can be seen that the boundary area of the transfer droplet was defective, which would have led to the deviation of the transferred droplet volume. Hence, the error compensation method was further applied. The repaired 3D export model is shown in Figure 9c. It can be seen that the surface of the transferred droplet was relatively complete without obvious defects.

It can be seen from Figure 9c that the shape of the transfer droplet was not completely regular; thus, the most accurate method to determine the volume of the transfer droplet was to take the coordinate point as calculation unit. Firstly, each calculation unit was integrated, and then all the integration results were added. After error compensation processing, nearly 160,000 calculation units needed to be integrated and summed, requiring a large amount of calculation. Therefore, the integral summation function was used to calculate the volume of the transfer droplet, and the measured volume of this transfer droplet was finally obtained as 28.4 pL. The volume of the remaining transfer droplets was measured using the above method. The error between the measured volume and the theoretical volume is shown in Figure 10a. We found that the measured volume was smaller than the theoretical volume. The average measured volume was 30.4 fL, which is about 96.4% of the theoretical average volume. Two reasons could explain why the measured volume was smaller than the theoretical volume. Firstly, the scan could have had missing parts, whereby error compensation processing would have only supplemented the missing parts with an approximate value close to the actual value. Secondly, the cross-section where the diameter of the transfer droplet was measured could have differed, as shown in Figure 10b. We found that the root of the actual transfer droplet was internal concave. Thus, the internal concave phenomenon became more obvious upon approaching the base plane, which would have made the measured volume smaller than the theoretical volume. After calculation, the average relative error of the transfer droplet volume was 3.8%. After analysis, we found that the theoretical calculation method was accurate, and the level of calculation was small. Therefore, the theoretical calculation method was used to estimate the transfer droplet volume in this paper.

### 4.2. Analysis of Influencing Factors

According to the conclusion drawn in Section 2.2, we could simplify the research content of this paper by studying the influence of the transfer droplet radius and pipetting needle diameter on the adhesive transfer rate. It can be seen from Equation (4) that the radius of the transfer droplet is the main factor affecting the volume of the transfer droplet. Through observation, we found that the stay time, initial distance, and stretching speed all affected the radius of the transfer droplet. The diameter of the pipetting needle mainly affects the volume of the initial droplet, and the volume of the initial droplet affects the volume of the transfer droplet. Without affecting the experimental results, we could simplify the research content of this paper by studying the influence of the initial droplet volume, stay time, initial distance, and stretching speed on the transfer droplet volume.

The four main influencing factors are defined below. The initial droplet volume refers to the volume of the droplet formed after the tip of the pipetting needle adheres to the adhesive. The stay time refers to the time that the pipetting needle tip remains stationary after contacting the base surface. The initial distance refers to the distance from the tip of the pipetting needle to the base surface. The stretching speed refers to the average speed of stretching the liquid bridge until fracture. Assuming a lack of interaction between factors, the single-factor method was used to analyze the influence of each factor on the transfer droplet volume.

#### 4.2.1. The Effect of Initial Droplet Volume

In the experiment, with other factors unchanged, initial droplets of different diameters were obtained by selecting pipetting needles of different diameters to attach the adhesive. The effect of initial droplet volume on transfer droplet volume was analyzed in detail through experiments. The stay time was set to 1 s, the initial distance was set to 0 μm, the stretching speed was set to 4 mm/s, and the pipetting needle diameters were 20 μm, 40 μm, 60 μm, and 80 μm. Figure 11 shows the transfer droplets observed under the microscope. It can be seen from the figure that the morphology of all transferred droplets was intact without obvious defects.

In order to ensure the reliability of the experimental results, after four groups of dispensing experiments were completed with pipetting needles of different diameters, 10 transfer droplets were randomly selected from each group for the measurement of diameter. The measurement results are shown in Figure 12a. It can be seen from the figure that the diameter of the same group of transfer droplets was approximately the same. Figure 12b reflects the relationship between the average diameter of the transfer droplet and the diameter of the pipetting needle. It can be seen from the figure that the transfer droplet diameter increased with the increase in the pipetting needle diameter. The initial droplet volume and the transfer droplet volume were calculated using the method of theoretical calculation, and the adhesive transfer rate was further calculated on this basis. The results are shown in Figure 12c. The results show that the transfer droplet volume and the adhesive transfer rate increased with the increase in the initial droplet volume.

A larger diameter of the pipetting needle resulted in a larger initial droplet volume adhering to the tip of the pipetting needle. When the height of the liquid bridge remained unchanged, a larger initial droplet volume resulted in a larger contact area between the liquid bridge and the base surface, as well as a larger diameter of the transferred droplet. As the contact area increased, the adhesion effect of the base surface to the liquid bridge increased, adsorbing more adhesive, and the volume of the transfer droplet increased. Viscous force plays a leading role in the transition regime [16], and large initial droplets exert greater pressure on the base surface. Therefore, the adhesive transfer rate increased under the combined action of viscous force and pressure. Zhu’s team also reached the same conclusion [27], where the contact area was also mentioned as the main factor enhancing the bonding effect [17,18].

#### 4.2.2. The Effect of Stay Time

Four groups of experiments were conducted for the stay time, with each group only changing the stay time of the pipetting needle on the base surface, while other conditions remain unchanged to study the effect of stay time on the transfer droplet volume. A pipetting needle with a diameter of 60 μm was selected, the initial distance was set to 0 μm, the stretching speed was set to 4 mm/s, and the stay time was 1 s, 2 s, 3 s, and 4 s. After the experiment, the bottom area of the transfer droplet was measured, and the micro-morphology of the transfer droplet was observed. The microscopic morphology of the transfer droplets formed under different stay times is shown in Figure 13. It can be seen from the figure that the bottom area of the transfer droplet increased with the increase in stay time. The microscopic morphology of the transfer droplet did not change due to the change in stay time.

Ten transfer droplets were randomly selected from the groups for bottom area measurement, and the measurement results of transfer droplet bottom area are shown in Figure 14a. It can be seen that changing the stay time of the pipetting needle on the base surface changed the bottom area of the transfer droplet, and the bottom area of transfer droplets obtained from the same group of experiments also fluctuated within a certain range. The average value of the transfer droplet diameter from each group was calculated, and the results are shown in Figure 14b. It can be seen more intuitively from the figure that there was a linear relationship between the transfer droplet diameter and the stay time. We found that a longer stay time led to a larger diameter of the transfer droplet. Upon only changing the stay time of the pipetting needle on the base surface, it was found that the volume of the initial droplet changed little. With the increase in stay time, the transfer droplet volume and the adhesive transfer rate increased. The specific values are shown in Figure 14c.

In order to ensure the quality of adhesive transfer, it is usually required that the adhesive remains relatively stationary for several seconds after contacting the base surface. In a relatively static state, the speed of the pipetting needle and the base surface acting on the liquid bridge tends to zero, and the transfer state of the adhesive belongs to a quasi-static state. Therefore, we can study the relationship between the stay time and the transfer rate by studying the transfer state of the adhesive in the quasi-static state, where surface tension plays an important role [16]. With the increase in residence time, the adhesive promoted the liquid bridge to gradually reach an equilibrium state under the action of mass force, resulting in an increase in the pressure of the liquid bridge on the base surface. The surface tension decreased with increasing pressure [28]; thus, an increase in the contact area between the liquid bridge and the base surface occurred. The increase in contact area meant that there was a smaller receding contact angle between the liquid bridge and the base surface, whereby the attraction of the base surface to the adhesive was enhanced. With a stronger attraction of the base surface to the adhesive, more adhesive was transferred to the base surface, forming larger transfer droplets and increasing the transfer rate. This conclusion is consistent with the conclusion obtained in [28], proving the correctness of the experiment in this paper.

#### 4.2.3. The Effect of the Initial Distance

Keeping the pipetting needle diameter, stay time, and stretching speed unchanged, different initial distances were set to study the effect of the initial distance on the volume of the transfer droplet. A pipetting needle with a diameter of 60 μm was selected, the stay time was set to 1 s, the stretching speed was set to 4 mm/s, and the initial distances were 0 µm, 6 µm, 12 µm, and 18 µm. The microscopic morphology of the transfer droplets formed under different initial distances is shown in Figure 15. It can be seen from the figure that transfer droplets of different sizes could be obtained by setting different initial distances.

Ten transfer droplets were randomly selected from the groups for diameter measurement, and the measurement results of the diameter of transfer droplet are shown in Figure 16a. It can be seen from the figure that varying the initial distance altered the diameter of the transfer droplet. A smaller initial distance resulted in a larger transfer droplet average diameter. This result is shown in Figure 16b. Through further analysis, we found that changing the initial distance had little effect on the initial droplet volume, but an increase in the initial distance led to a decrease in the transfer droplet volume and adhesive transfer rate. This result is shown in Figure 16c.

Without changing any configuration parameters, the initial droplet with almost the same volume could be obtained each time. When the initial droplet contacted the base surface and formed a liquid bridge, the volume of the liquid bridge became the initial droplet volume. As the initial distance increased, the liquid bridge became finer and longer, and the contact area between the adhesive and the base surface became smaller. Therefore, a larger initial distance resulted in a smaller diameter of the transfer droplet. The research results in [29] showed that the transfer rate is related to the viscous force. As the initial distance increased, the viscous force between the liquid bridge and the base surface gradually weakened, whereby the attraction of the base surface to the adhesive became smaller. Therefore, with the initial droplet volume almost constant, the initial distance increased, the transfer droplet volume decreased, and the adhesive transfer rate decreased. In this paper, the same conclusion as that of [29] was also obtained through experiments, indicating that the adhesive transfer rate under microscopic and macroscopic conditions had the same law and obeyed the requirements of conventional hydrodynamics.

#### 4.2.4. The Effect of Stretching Speed

The stretching speed of the pipetting needle was varied, with other factors remaining unchanged, to analyze the relationship between the stretching speed and the volume of the transfer droplet. A pipetting needle with a diameter of 60 μm was selected, the stay time was set to 1 s, the initial distance was set to 0 μm, and the stretching speed was divided into seven groups (1–7 mm/s). The microscopic morphology of the transfer droplets formed under different stretching speeds is shown in Figure 17. It can be seen from the figure that changing the stretching speed had an effect on the size of the transfer droplets. However, the stretching speed had almost no effect on the morphology of transfer droplets, and a complete transfer droplet could be obtained at any stretching speed.

Ten transfer droplets were randomly selected from the groups for diameter measurement, and the results are shown in Figure 18a. It can be seen from the figure that the diameter of the transfer droplet changed with the change in stretching speed. According to the relationship between the average diameter of the transfer droplet and the stretching speed shown in Figure 18b, it can be seen that a faster stretching speed led to a larger diameter of the transfer droplet. Changing the stretching speed had almost no effect on the volume of initial droplet; however, with the increase in stretching speed, the volume of the transfer droplet and adhesive transfer rate decreased. The specific values are shown in Figure 18c.

The above research results are consistent with [30]. The phenomenon of the transfer droplet diameter increasing with the increase in stretching speed can be explained according to two aspects. Firstly, a smaller stretching speed resulted in a smaller velocity gradient in the vertical direction inside the adhesive. With an increase in stretching speed, the velocity gradient resulted in the two ends of the liquid bridge being very small while the middle was very large, leading to slow momentum propagation in the middle. Secondly, when different stretching speeds were used to stretch the liquid bridge to the same height, a faster stretching speed led to a shorter stretching time. Accordingly, at a faster stretching speed, the degree of inward sliding of the contact line of the base surface decreased, and the transfer droplet diameter increased.

The main reasons for the decrease in the adhesive transfer rate could be explained by the surface tension playing a leading role in the transfer process when the stretching speed was low. The main factor affecting surface tension is the receding contact angle. A smaller receding contact angle resulted in a greater attraction to the adhesive. Because the receding contact angle of the base surface was smaller than the pipetting needle, the base surface was more attractive to the adhesive, and more adhesive was transferred to the base surface. With an increase in stretching speed, the velocity gradient in the vertical direction of the liquid bridge increased, resulting in a slower momentum propagation of the adhesive and a lower adhesive transfer rate.

### 4.3. The Experiment of fL–pL Level Automatic Dispensing

#### 4.3.1. Adhesive Automatic Distribution of fL–pL Level

Through the single-factor experiment, it was found that the initial droplet volume, stay time, initial distance, and stretching speed all affected the transfer droplet volume. By calculating the overall standard deviation, it was found that, relative to other levels, a diameter of the pipetting needle of 20 μm, a stay time of 3 s, an initial distance of 0 μm, and a stretching speed of 2 mm/s were optimal to obtain the transfer droplet with the best uniformity.

According to the design requirements, the movement trajectory was planned. Figure 19 shows a comparison diagram of the automatic dispensing effect. The preset distance between two adjacent vertices of the quadrangular star was 1600 μm, and the actual distances were 1607 μm, 1609 μm, 1610 μm, and 1609 μm. The difference rates were 0.4%, 0.6%, 0.6%, and 0.6%, respectively. The preset circle diameter was 2200 μm, and the preset distance between adjacent transfer droplets in the circle was 275 μm. The actual values were 2191 μm and 272 μm, with difference rates of 0.4% and 1.1%, respectively.

In order to reflect the difference between the planned trajectory and the actual trajectory, the planned position of each transfer droplet in the quadrangular star was compared with the actual position. The compound effect of planned location and actual location is shown in Figure 20. It can be seen from the figure that the actual transfer droplet trajectory was basically consistent with the preset transfer droplet trajectory. After comparison, four representative transfer droplets were selected. Among them, transfer droplet 1 and transfer droplet 2 had good morphology and position accuracy, while the morphology of transfer droplet 3 was intermediate, and that of transfer droplet 4 was poor. After the transfer droplets were compounded, the diameter and position deviation of the transfer droplets were measured, and the deviation rate was calculated. After measurement, it was found that the minimum diameter of the transfer droplet was 23.5 μm, the maximum diameter was 24.5 μm, and the deviation between the planned position and the actual position was 0.6–2.6 μm. After calculation, the average diameter of the transfer droplets was obtained as 23.9 μm, the average deviation was 1.5 μm, the deviation rate was 2.5–10.8%, and the average deviation rate was 6.2%. The reason for the large position deviation of some transfer droplets could be that, during the dispensing process, the driving device had a slight vibration relative to the base surface, which caused the actual position of the transfer droplet to deviate.

It can be seen from Section 4.1 that the theoretical value of the volume of the transfer droplet was close to the measured value; thus, due to its simplicity, it was used to calculate the volume of the transfer droplet. After measurement, it was found that the minimum height of the transfer droplet was 3.1 μm, and the maximum height was 3.6 μm. The minimum contact angle of the transfer droplet was 34°, and the maximum contact angle was 40°. The minimum radius of the transfer droplet was 11.7 μm, and the maximum radius was 12.3 μm. Specific data are shown in Figure 21a. After calculation, the minimum volume of the initial droplet was obtained as 1188.6 fL, the maximum volume was 1456.2 fL, and the average volume was 1303.5 fL. The minimum volume of the transfer droplet was obtained as 573.3 fL, the maximum volume was 779.9 fL, and the average transfer volume was 659.6 fL. The minimum transfer rate of adhesive was obtained as 46.1%, the maximum transfer rate is 55.6%, and the average transfer rate is 50.6%. Specific data are shown in Figure 21b.

Through the fL-level adhesive automatic distribution experiment, it could be found that the actual position was basically consistent with the preset position, revealing an average deviation of 6.2% and a minimum volume of the transfer droplet of 573.3 fL. Therefore, the ultra-micro-dispensing system designed in this paper could realize the automatic dispensing of adhesive at an fL level. Compared with the existing adhesive transfer system, this system has the advantages of better stability and a smaller transfer volume.

#### 4.3.2. The Application of Microporous Encapsulation

The internal structure of an MEMS is precise and complex. In the actual assembly process, ultra-micro-dispensing technology is often used to connect and encapsulate elements of the microelectromechanical system. Microporous encapsulation faces problems that need to be solved. Because the encapsulation experiment needs to be carried out in micron-level small holes, common dispensing methods cannot ensure an accurate position and appropriate adhesive filling volume. Therefore, it is necessary to design a new distribution mechanism to solve the problem of microporous encapsulation.

The automatic dispensing method designed in this paper can realize microporous encapsulation. During the adhesive filling process, the pipetting needle should not be in contact with the micropore; therefore, a pipetting needle with a diameter of 200 μm was selected for adhesive filling, the initial distance was set to 2 μm, and the rise amount was set to 1 μm. Under the action of viscous force, the adhesive was transferred to the micropore. When the adhesive was observed to overflow the micropore, the microporous encapsulation was completed. The effect of microporous encapsulation is shown in Figure 22. It can be seen from the figure that, after the microporous encapsulation experiment, the microporous structure was complete, and the adhesive could be used to evenly fill the micropore. Therefore, the ultra-micro-automatic dispensing method based on surface tension designed in this paper could meet the actual encapsulation needs.

## 5. Conclusions

According to the principle of liquid transfer printing, this paper studied the transfer mechanism of adhesive from a capillary tube to the base surface using a pipetting needle, and an ultra-micro-dispensing method was proposed. This method could realize the automatic distribution of adhesive with fL–pL resolution.

Firstly, the theoretical value of the transfer droplet volume was calculated, and then the actual value of the transfer droplet volume was measured using 3D scanning. The results showed that the measured volume was about 96.4% of the theoretical volume.

By studying the effects of initial droplet volume, stay time, initial distance, and stretching speed on the transfer droplet volume, it was found that the transfer droplet volume increased with the increase in the initial droplet volume, stay time, and stretching speed, whereas it decreased with the increase in initial distance. The adhesive transfer rate was also affected by these four factors, and the influence rule was consistent with the transfer droplet volume. The initial droplet volume increased with the increase in the pipetting needle diameter, whereas the stay time, initial distance, and stretching speed had little effect on the initial droplet volume.

By planning the movement trajectory, the automatic dispensing of transfer droplets with fL resolution in the plane was realized. After calculation and analysis, the dispensing results were as follows: a minimum radius of the transfer droplet of 11.7 μm, a minimum transfer volume of 573.3 fL, an average transfer volume of 659.6 fL, and an average transfer rate of 50.6%. The average deviation of the automatic dispensing position was 1.5 μm. In addition, the encapsulation experiment using a Φ225 μm × 70 μm micropore was successfully completed, further indicating that the ultra-micro-volume dispensing method proposed in this paper can be applied to the encapsulation of micro-sized parts.

## Figures and Tables

**Figure 1 micromachines-13-00664-f001:**
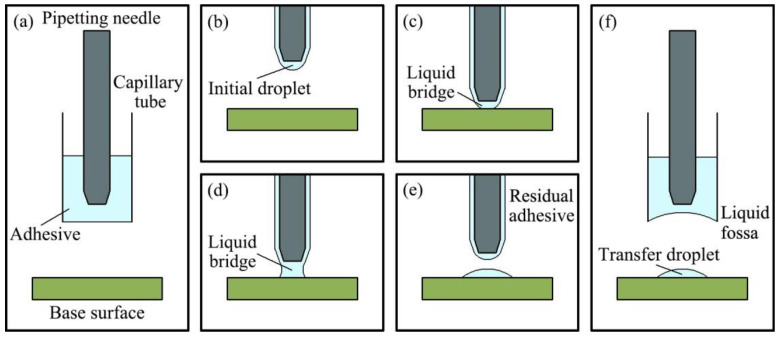
Diagram illustrating the principle of the ultra-micro-dispensing process: (**a**) adhesive taking; (**b**) transportation; (**c**) extrusion; (**d**,**e**) stretch and break; (**f**) return.

**Figure 2 micromachines-13-00664-f002:**
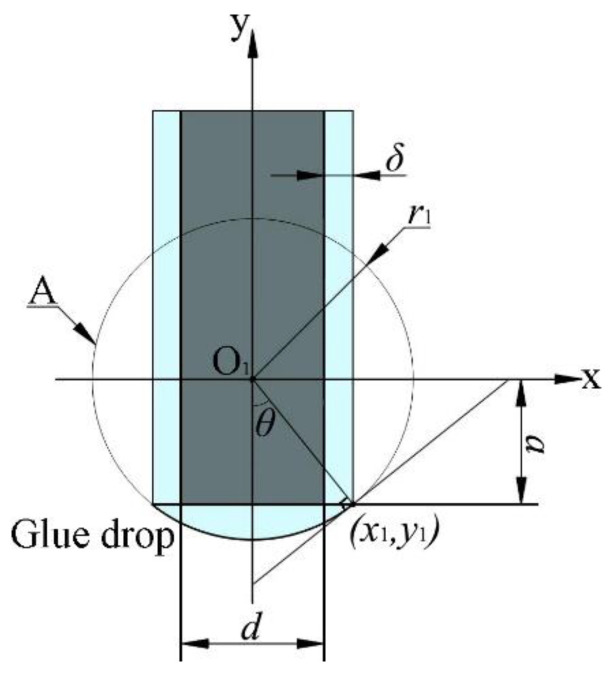
Ideal initial droplet model.

**Figure 3 micromachines-13-00664-f003:**
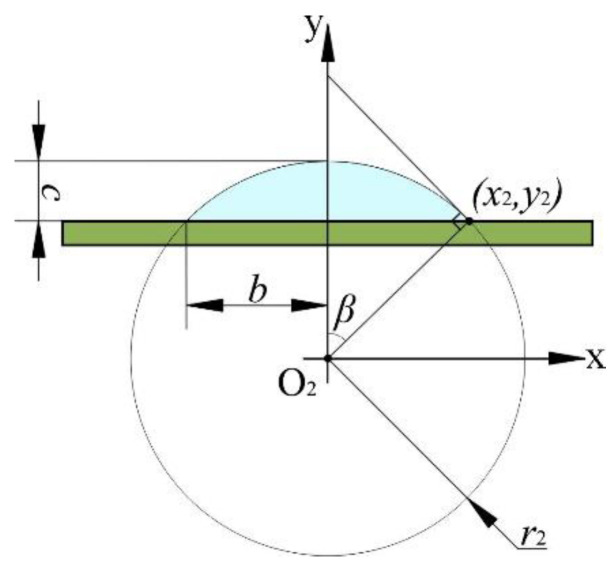
Ideal transfer droplet model.

**Figure 4 micromachines-13-00664-f004:**
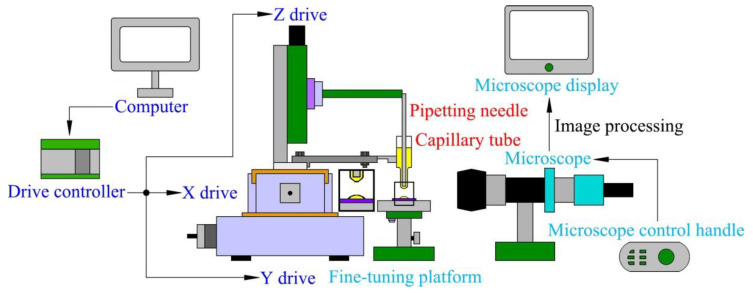
Composition diagram of automatic dispensing system.

**Figure 5 micromachines-13-00664-f005:**
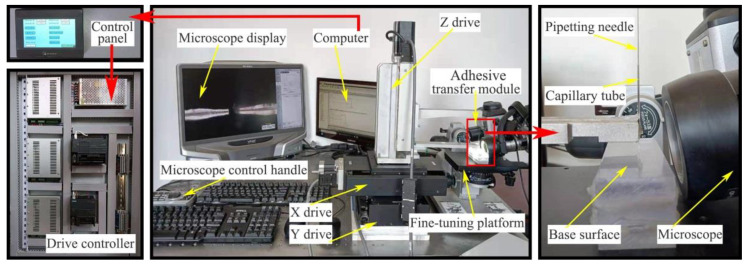
Physical object of experimental platform.

**Figure 6 micromachines-13-00664-f006:**
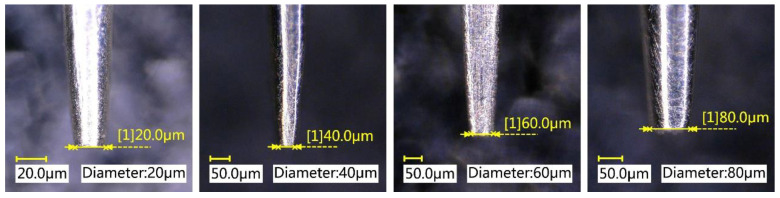
The pipetting needle image.

**Figure 7 micromachines-13-00664-f007:**
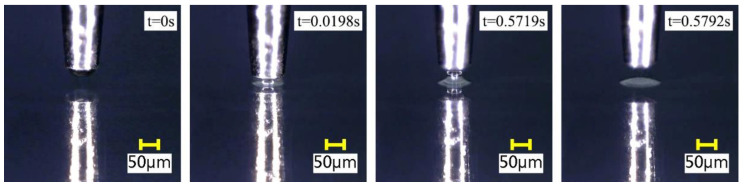
The actual transfer process of the adhesive.

**Figure 8 micromachines-13-00664-f008:**
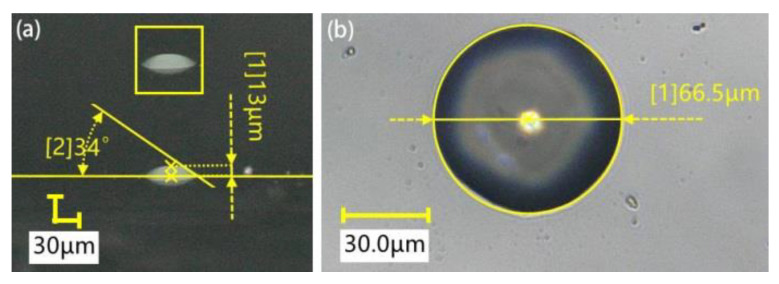
Transfer droplet parameters: (**a**) horizontal direction; (**b**) vertical direction.

**Figure 9 micromachines-13-00664-f009:**
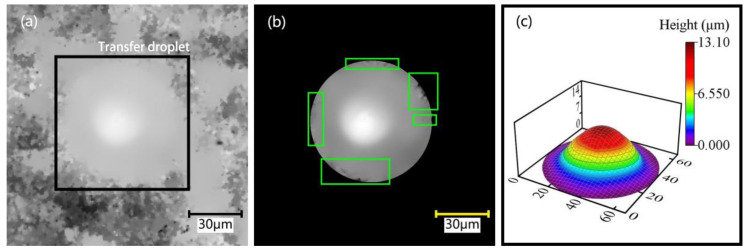
The 3D scanned image: (**a**) original image; (**b**) processed image; (**c**) 3D exported model.

**Figure 10 micromachines-13-00664-f010:**
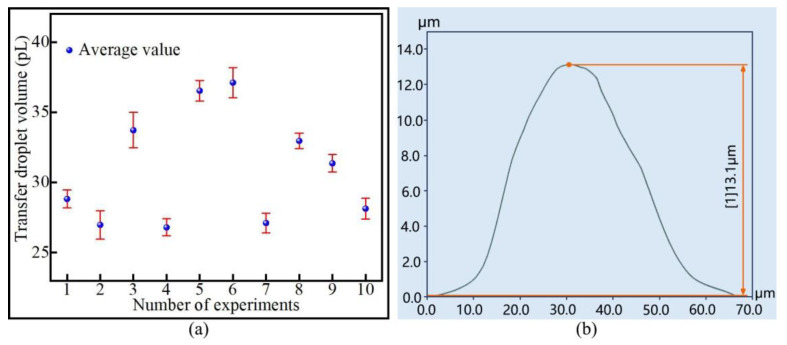
Transfer droplet volume: (**a**) comparison between the measured volume and the theoretical volume; (**b**) cross-section profile of the transfer droplet.

**Figure 11 micromachines-13-00664-f011:**
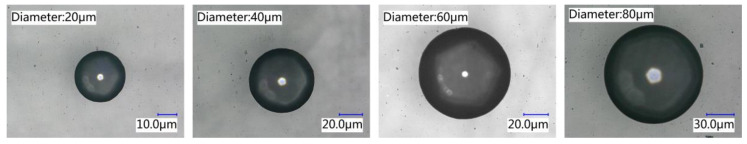
Transfer droplets obtained after changing the diameter of the pipetting needle.

**Figure 12 micromachines-13-00664-f012:**
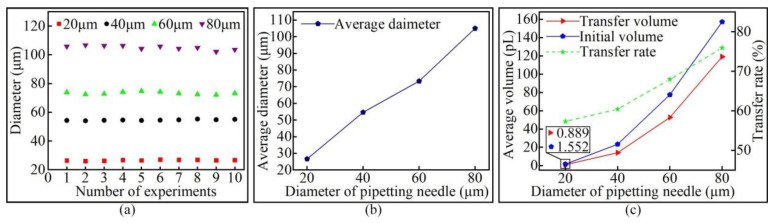
The relationship between the initial droplet volume and the transfer droplet volume: (**a**) the diameter of the transfer droplet; (**b**) the average diameter of the transfer droplet; (**c**) volume comparison.

**Figure 13 micromachines-13-00664-f013:**
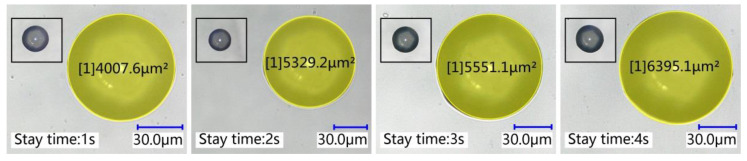
The transfer droplet obtained after changing the stay time.

**Figure 14 micromachines-13-00664-f014:**
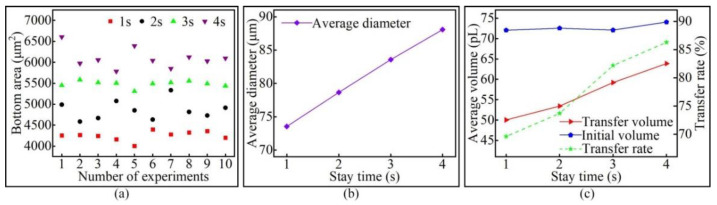
The relationship between the stay time and the transfer droplet volume: (**a**) the bottom area of the transfer droplet; (**b**) the average diameter of the transfer droplet; (**c**) volume comparison.

**Figure 15 micromachines-13-00664-f015:**
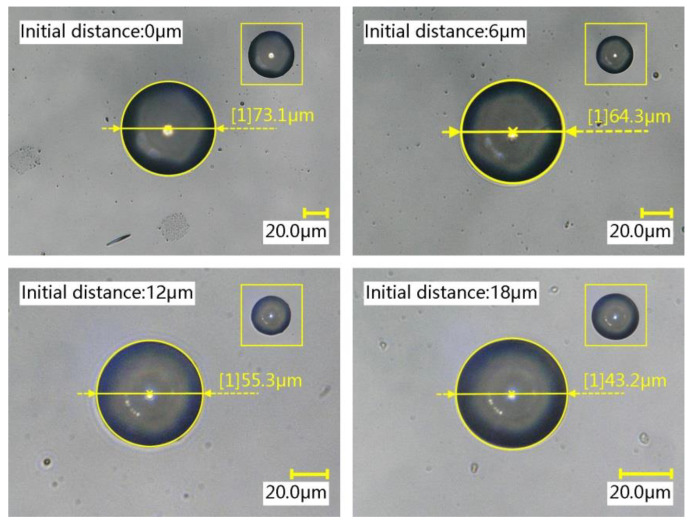
The transfer droplets obtained after changing the initial distance.

**Figure 16 micromachines-13-00664-f016:**
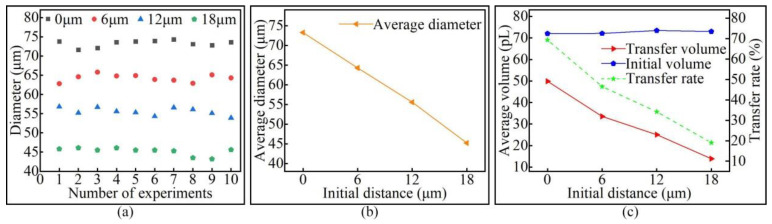
The relationship between the initial distance and the transfer droplet volume: (**a**) the diameter of the transfer droplet; (**b**) the average diameter of the transfer droplet; (**c**) volume comparison.

**Figure 17 micromachines-13-00664-f017:**
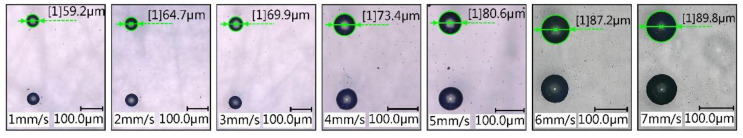
The transfer droplet obtained after changing the stretching speed.

**Figure 18 micromachines-13-00664-f018:**
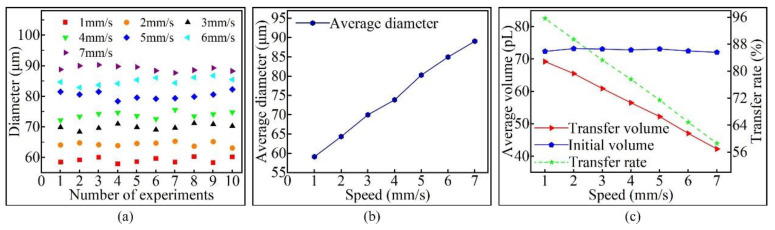
The relationship between the stretching speed and the transfer droplet volume: (**a**) the diameter of the transfer droplet; (**b**) the average diameter of the transfer droplet; (**c**) volume comparison.

**Figure 19 micromachines-13-00664-f019:**
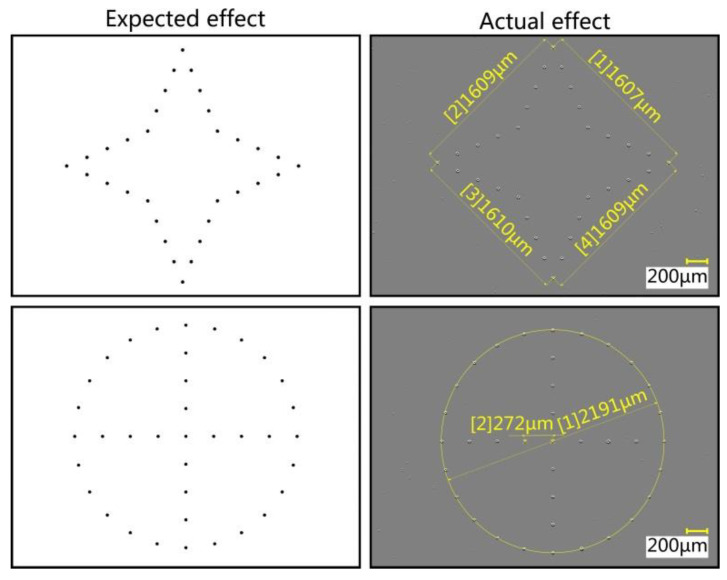
Comparison diagram of automatic dispensing effect.

**Figure 20 micromachines-13-00664-f020:**
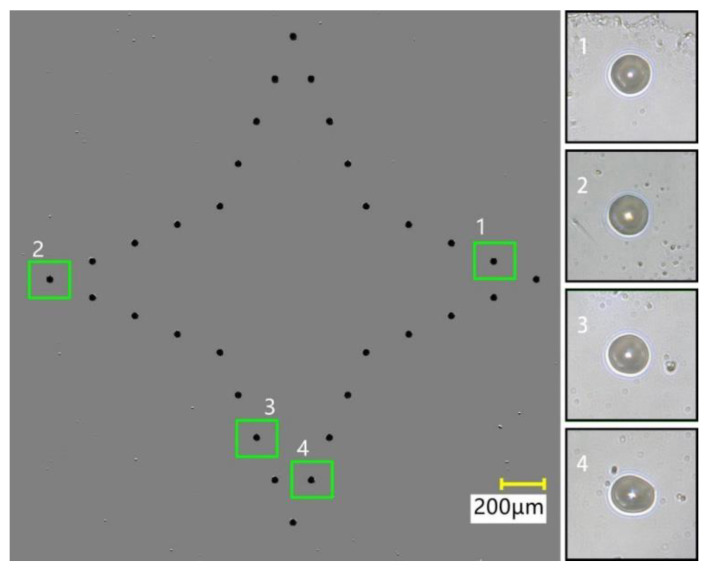
Compound effect comparison.

**Figure 21 micromachines-13-00664-f021:**
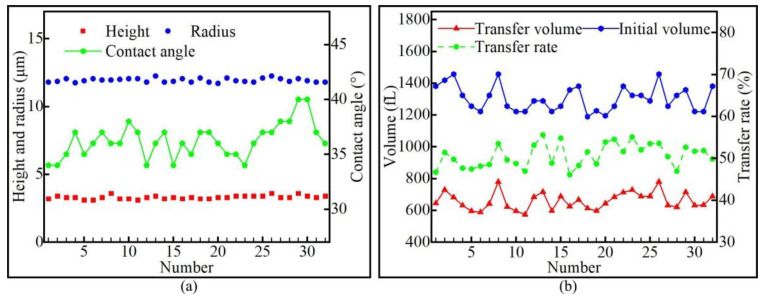
Quadrangular star parameter: (**a**) transfer droplet parameter; (**b**) adhesive transfer rate.

**Figure 22 micromachines-13-00664-f022:**
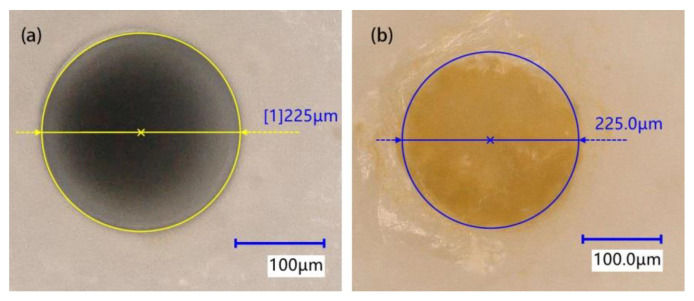
Microporous encapsulation effect: (**a**) before encapsulation; (**b**) after encapsulation.

**Table 1 micromachines-13-00664-t001:** Measurement data and calculation results.

Number	1	2	3	4	5	6	7	8	9	10
*c* (μm)	13	13	14	13	15	15	13	14	14	13
*b* (μm)	33.3	32.4	33.2	31.9	32.1	32.8	33.1	32.9	32.3	32.6
*β* (°)	34	35	33	35	34	34	36	34	35	34
*V*_2_ (pL)	29.3	27.7	34.7	27.2	37	37.9	27.6	33.4	31.8	28.7

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
