# Peer review of "An Ultra-Micro-Volume Adhesive Transfer Method and Its Application in fL–pL-Level Adhesive Distribution"

_micromachines, 2022, doi:10.3390/mi13050664_

Round 1

Reviewer 1 Report

In this manuscript, the authors introduced an ultra-micro dispensing method and realized the automatic distribution of adhesive with fL-pL resolution. They explored the transfer mechanism of the adhesive from the capillary tube to the base surface via pipetting needle. The general scope of the paper is well justified and of interest to a sufficiently broad audience. The main finding of this paper appears to be that the transfer droplet volume was determined by the initial droplet volume and the stay time, but not by the initial distance and stretching speed. In general, this work provides an effective way to obtain the automatic distribution of adhesive at an extremely-low scale. Although this idea is original and would be beneficial for microporous encapsulation in MEMS systems, the authors should consider the following points in any form of revision.

  1. During the process of the ultra-micro dispensing, from figure 1d to figure 1e, the transferred adhesive formed transfer droplets under the surface tension and viscous force. However, a question cropped up in my mind whether there will be microdroplet formation during the liquid bridge fracture caused by Rayleigh instability.
  2. To the best of my knowledge, the final size of the adhesive microdroplet formed on the base surface largely depends on the size of the contact line at the solid-liquid interface. Therefore, the wettability of the base surface is undoubtedly a critical factor during this process. The authors should add more discussion to this part.
  3. Page 7, Figure 6 and Figure 7, time scale should be added in the process of adhesive transfer.
  4. Page 2, line 71-74, “Aiming at the common dispensing technology is difficult to meet the ultra-micro volume demand of micro nanometer manufacturing for adhesive distribution technology, the influence of viscous force and surface tension on dispensing process in micro nanometer scale, and the problem of small nozzle manufacturing difficult.....”, the sentence was confusing.
  5. At the end of page 4, line 152, “he contact angle...” should be the contact angle...

Overall, I recommend this paper be published in Micromachines after some minor revisions.

Reviewer 2 Report

  1. For the introduction, I would suggest to elaborate more on conventional technologies currently used for adhesive transfer such dispensing, jetting and others. To include some examples, such a minimum droplet diameter achieved by dispensing and jetting.
  2. You also mentioned that “Therefore, it is widely used in the assembly of micro systems, such as micro gyroscope, micro accelerometer and so on [6]”. I am suggesting to give examples of specific features on mentioned above devices there and how you developed technology can help to solve that.
  3. To illustrate you findings you used “epoxy resin with a viscosity of 1000cps is selected as the adhesive”. It is OK.

Why you use specifically such material? Why it must have a viscosity of 1000cps?

What will happened if you use more viscous and less viscous adhesive?

Did you method have upper or low viscosity limit?

  1. Does you method applicable specifically for the epoxy adhesive? Is it suitable for silicone-based adhesive such PDMS and etc.?
